# A New Transmission Route for the Propagation of the SARS-CoV-2 Coronavirus

**DOI:** 10.3390/biology10010010

**Published:** 2020-12-26

**Authors:** Antoine Danchin, Tuen Wai Ng, Gabriel Turinici

**Affiliations:** 1Institut Cochin, INSERM U1016-CNRS UMR8104-Université de Paris, 75006 Paris, France; antoine.danchin@normalesup.org; 2School of Biomedical Sciences, Li Ka Shing Faculty of Medicine, The University of Hong Kong, Pokfulam, Hong Kong SAR, China; 3Department of Mathematics, The University of Hong Kong, Pokfulam, Hong Kong SAR, China; 4CEREMADE UMR 7434 CNRS, Université Paris Dauphine-PSL, 75016 Paris, France; gabriel.turinici@dauphine.fr

**Keywords:** COVID-19, SARS-CoV-2, epidemic model, contamination, SARS, coronavirus, coronavirus propagation

## Abstract

**Simple Summary:**

In animals, two dominant organs are infected by coronaviruses, the respiratory tract and the gut. In past epidemics, a shift from the lung to the gut or vice versa has been observed. Analysis of the SARS episode in 2003 had suggested the possibility of a “double epidemic” explaining the absence of severe cases in Shanghai at the time. Here, using data from Wuhan and Hong Kong, Singapore, and Shenzhen, we discuss whether the present respiratory contagion route for SARS-CoV-2 might have also resulted from faecal contamination. We noticed that the propagation of the disease in the city of Wuhan underwent an original course. Besides the expected respiratory route, the disease appeared to have been greatly facilitated by a secondary propagation route, thus substantiating the beneficial effect of an effective quarantine. By contrast, a similar situation based on early data did not prevail in Hong Kong or Singapore. Our work shows how health authorities could decide to orient their prevention measures, depending on the analysis of ongoing epidemics, as a function of the identification of a secondary route in parallel with the prevailing one. Our model could also help highlight voluntarily or accidentally biased data collection.

**Abstract:**

Background: Starting late 2019, a novel coronavirus spread from the capital of the Hubei province in China to the rest of the country, then to most of the world. To anticipate future trends in the development of the pandemic, we explore here, based on public records of infected persons, how variation in the virus tropism could end up in different patterns, warranting a specific strategy to handle the epidemic. Methods: We use a compartmental model to describe the evolution of an individual through several possible states: susceptible, infected, alternative infection, detected, and removed. We fit the parameters of the model to the existing data, taking into account significant quarantine changes where necessary. Results: The model indicates that Wuhan quarantine measures were effective, but that alternative virus forms and a second propagation route are compatible with available data. For the Hong Kong, Singapore, and Shenzhen regions, the secondary route does not seem to be active. Conclusions: Hypotheses of an alternative infection tropism (the gut tropism) and a secondary propagation route are discussed using a model fitted by the available data. Corresponding prevention measures that take into account both routes should be implemented to the benefit of epidemic control.

## 1. Background

Late in 2019, a novel coronavirus was detected in mainland China originating from Wuhan, the capital of the Hubei province. The importance of the disease took some time to be acknowledged [1,2], resulting in a fairly significant number of infected persons who subsequently spread the disease throughout China [3] (all provinces had at least one case on 30 January 2020), then worldwide. This makes it essential to explore the way the epidemic may spread in the future. In the present work, we propose several scenarios to this aim, centered around an alternative propagation route. While it is obviously hazardous to advance models of epidemics before their course has been completely unfolded, we think that it is helpful to evaluate methods meant to understand their specificity. In particular, by following how the epidemic developed at places other than its original site of onset, we would be able to detect any unexpected development course that might be used by health authorities to react. This would be extremely useful to detect patterns created by socio-political measures meant to contain the disease or mutants of the virus which would result in higher contagion or virulence, thus prompting a rapid response.

It has been established that coronaviruses are versatile in their preferred site of infection. These viruses have the option to pass from a “gut tropism” version to a “lung tropism“ instance, as was observed for other coronavirus outbreaks [4,5,6,7]. Depending on the infected person or virus spread dynamics, coronavirus effects can be more or less severe: the virus can either preserve its lung tropism correlated with high impact, or act as a “gut tropism” version and be relatively harmless (or even asymptomatic). First reports established that SARS-CoV-2 also induces gut symptoms [8]. Its genetic build-up and evolution is still subject to intense research [9]. This multifaceted behaviour may result in unexpected local courses of the epidemic, as suggested in a scenario of a “double epidemic”, and as proposed for the SARS 2003 worldwide epidemic [10], except that in the present situation we would be witnessing the effect of a single epidemic with modulated effects and propagation, depending on the affected patients and random virus mutations. The versatility of the virus tropism, in addition to interfering with the immune response, may induce an additional propagation route depending on virus tropism; contagion may involve a variety of causes, such as environments contaminated with virus carrier secretions, dirty water effluents, besides the expected direct contamination via aerosols. This means that the orofecal route should be considered as an important complement of contact with the virus (see [11] for recent information that sustains this hypothesis).

Coronavirus-mediated diseases have been frequent in domestic animals for a very long time. They were associated to two major organs, the gut and the lung. There is a direct link between the lung and gut because lung secretions are continuously swallowed by animals, man included. The symmetrical route is open when fecal matter reaches the oral cavity, which is frequently open for respiration. A severe epidemic affecting pigs almost four decades ago, a shift of tropism from the gut to the lung, changed the course of the disease [12]. During the SARS episode, it was observed (e.g., at the Amoy Garden cluster of cases in Hong Kong) that SARS-CoV-1 could be transmitted via sewage [13,14,15]. This route has also been demonstrated with SARS-CoV-2 [16]. This is likely the consequence of aerosols associated to sewage or of soiled hands reaching the oral cavity or the nose; hence, the importance of repeatedly washing hands.

This observation prompted us to include in our model, besides the major respiratory route, an additional propagation route, which is not a direct human-to-human propagation but invokes some indirect, vector, or environment element, to/from humans. A second important consequence of assuming the presence of an indirect route is that the selection pressure on virus mutants will differ considerably between lung tropism and gut tropism.

When a virus reaches its target cell it begins to multiply. This draws on the cell’s metabolic resources. While the general outline of the metabolism is the same in different hosts, there may be considerable variation from person to person, depending on its environment or social status, such as its age or previous infections. As a consequence, this will impact the selection pressure on the virus itself (it is considerably dependent on nucleotide metabolism, for example, see [17]) and this will be reflected in the incubation period (i.e., the delay necessary for a virus to yield visible consequences in a host), which will depend on the virus reproduction rate, with a low rate leading to a longer incubation period.

This hypothesis is consistent with some propagation patterns, such as the first case in Macau, which was not detected at the border, implying that the affected person did not have fever. We should note that this makes the disease considerably more dangerous in terms of propagation because carriers are, at least for some time, “invisible” and display risky behavior [18,19]. This explains our choice of compartmental model (see below).

Very recent clinical and epidemiological evidence sustaining our hypothesis of alternative route of propagation can be found in [11,16,20,21].

We propose several scenarios of coronavirus propagation in parallel with our propagation model. Besides providing suggestions to the operational epidemic propagation models, this work also suggests some countermeasures to slow down propagation. Compared with SARS and the flu, the overall burden became much higher among others because of lack of proper containment.

## 2. Methods

The mathematical model of epidemic propagation is illustrated in Figure 1; it builds on a compartmental model [22,23] to describe the evolution of an individual through several possible conditions, such as susceptible (compartment label “S”), latent, that is, infected, but not yet infective (label “L”), infective in the lung tropism and highly symptomatic (label “I”), low symptomatic, infective in the gut tropism (“A”); the “I” infected then go to “Detected” (“D”) stage, while the “A” infected go to the “Removed /recovered” stage (label “R”). This is an adaptation of existing models and especially the SLIAR model [24,25,26,27,28]. We add to the model the possibility for an environmental, local, propagation route (that accounts for soiled water, etc.). This option is represented by the “E” label. With some probability, the contact between infected environmental elements and a susceptible individual gives rise to new infections.

Here, *N* is the total population; the parameters β, $\beta_{EI}$, $\beta_{ES}$, δ, $\gamma_{L}$, $\gamma_{I}$, $\gamma_{A}$, $\gamma_{E}$, L(0), I(0), A(0), R(0), E(0) and *p* of the model are fitted so that the “D” compartment matches the existing cumulative number of cases [29] known at the beginning of the epidemic. The mathematical formulation of the model is:(1)$$\begin{array}{ccc} & \; & dS/dt=-\beta S\left(I+\delta A\right)/N-\beta_{ES}ES \end{array}$$(2)$$\begin{array}{ccc} & \; & dL/dt=\beta S\left(I+\delta A\right)/N+\beta_{ES}ES-\gamma_{L}L \end{array}$$(3)$$\begin{array}{ccc} & \; & dI/dt=p\gamma_{L}L-\gamma_{I}I \end{array}$$(4)$$\begin{array}{ccc} & \; & dA/dt=\left(1-p\right)\gamma_{L}L-\gamma_{A}A \end{array}$$(5)$$\begin{array}{ccc} & \; & dD/dt=\gamma_{I}I \end{array}$$(6)$$\begin{array}{ccc} & \; & dR/dt=\gamma_{A}A \end{array}$$(7)$$\begin{array}{ccc} & \; & dE/dt=\beta_{EI}\left(I+\delta A\right)/N-\gamma_{E}E. \end{array}$$

Simulations start at some date $t_{0}$. The parameters were fit with a zero-order optimization procedure called *differential evolution* [30], as implemented in the Python “scipy” package. The procedure minimizes (without the need to compute any gradient) the following function: to any set of parameters of the model, we associate the least squares error, that is, the sum (over available calendar dates) of squares of differences between observed empirical values and the simulation result for the “D” class. We did no "cherry picking"—that is, we used the fitting parameters as they were output by the calibration process, without further selection; this is encouraging for the robustness of the model and the fitting procedure.

## 3. Results

We start with a simulation of the Wuhan city epidemic. We take into account the quarantine starting 23rd January and impose a reduction of the transmission parameters from nominal values β and $\beta_{ES}$ to reduced values $a_{1}\beta$ and $a_{2}\beta_{ES}$ post-quarantine start date. The reduction is represented by percentages $a_{1},a_{2}$ between 0% and 100% that are fitted numerically.

This attenuation is found, as is the case for all other parameters, in the course of a search procedure that imposes a match of the “Detected” patients predicted by the model and the number of reported cases available from communications of the WHO or Chinese authorities. The results are displayed in Figure 2. The epidemic size is around 28,500. The key observation here is the presence and importance of the alternative form “A” and the alternative propagation route E-S (see also below).

To exclude effects from late-epidemic stages, we only use data from the first months of propagation.

The effect and importance of quarantine measures: we tested what happens if we discarded the attenuation factors found by the fit procedure. We obtained a total epidemic size of 97,400, which is a sharp increase with respect to the baseline scenario. Note that these figures are not to be taken per se, but only as projections that can be made, at some point in time, with available data. Recall that the goal of the study is not to make propagation predictions, but to state whether the data available in the beginning of the epidemic are coherent with a second propagation route.

To demonstrate the importance of the alternative form and the alternative propagation route, we neglected the alternative propagation route by setting the $\beta_{ES}$ parameter to zero in the previous run. We obtained the results in Figure 3, which shows that absence of the E-S propagation route cannot explain the way the epidemic is unfolding (total epidemic size is down to 112 from previous figure of 28,500, closer to observations).

Similar considerations hold when switching down the alternative form of virus presentation (results not shown here).

A second simulation concerns the Hong Kong propagation (Figure 4): here, statistical uncertainties in the data do not allow a good fit, but the first wave epidemic appears to be contained; note that the model cannot foresee what can happen if new infected individuals turn up from outside or if quarantine efforts diminish in the model’s future.

We continue with a simulation of the propagation of the epidemic in Shenzhen (Figure 5): the epidemic seems to be relatively under control with a final epidemic size below 600 (although data were not regularly updated at the end of our selected time interval as the “Detected” class remains constant for a long while); this may force the fit procedure to overweight parameters that ensure sudden epidemic stop and thus a smaller total epidemic size. A sensitivity analysis (not shown here) can be used to produce a confidence interval for the final epidemic size.

Finally, we explored also the case of Singapore: the results in Figure 6 are similar to those of Hong Kong; there, the epidemic does not appear to be self-sustained.

## 4. Discussion

A word of caution concerning the present analysis—any model is but an imperfect view of reality, and this model is no exception. In addition, the quality of the results returned by the model depends directly on the quality of the data input, and it seems likely that some of the reported figures are subject to large error bars (especially the Wuhan city data). Therefore, one should take the exact figures in our simulations with a grain of salt. Yet, the simulation corresponding to the model do show important qualitative features that are now discussed. Specifically, the hypothesis that there exists a secondary route of transmission is validated by existing data while the assumption that two major types of virus tropism coexist is also supported by the simulation outputs (and was validated by clinical data). This has the interesting consequence that it appears that, for the city of Wuhan, the quarantine seems to be effective—of course notwithstanding the need for further quarantine efforts to ensure that it remains under control. However, since a reservoir is probably present, and the orofecal route may be an important propagation factor, the prevention of this transmission element is vital to avoid any resurgence. The first recommendation seems to be the enforcement of strict post-epidemic measures at the possible reservoir sites.

In Hong Kong and Singapore (and partially in Shenzhen as well), even if the current data indicate that the number of cases is likely to increase, it does not indicate that a secondary propagation route was already effective. However, efforts are to be made to ensure that this remains true in the future and the control of the secondary route, which makes the difference between a large scale epidemic and a controlled outbreak, remains an important target for public health measures.

## 5. Conclusions

Elaborating on the behavior of previous coronavirus outbreaks, we worked out the hypothesis that an alternative infection tropism (the gut tropism) linked to a secondary propagation route (through environment) is affecting the development of the present COVID-19 epidemic. Our epidemic propagation model, when fit to existing data, indicated that, among all regions analyzed (Wuhan city in the Hubei region in mainland China, Hong Kong, Singapore, Shenzhen), the propagation of the disease in the city of Wuhan underwent an original course. It appeared to be greatly facilitated by a secondary propagation route, thus substantiating the beneficial effect of an effective quarantine. The main message of our exploration is that relevant prevention measures that take into account both propagation routes are key to containing the extension of the epidemic to further sites, especially when novel sites are uncovered. However, it must be emphasized that because our work depends on figures provided by official health authorities, the scenarios we proposed could be considerably altered if the figures were not reflecting the real situation. In particular, the existence of a second, significant route might in fact provide a way to identify alterations of these figures.

## Figures and Tables

**Figure 1 biology-10-00010-f001:**
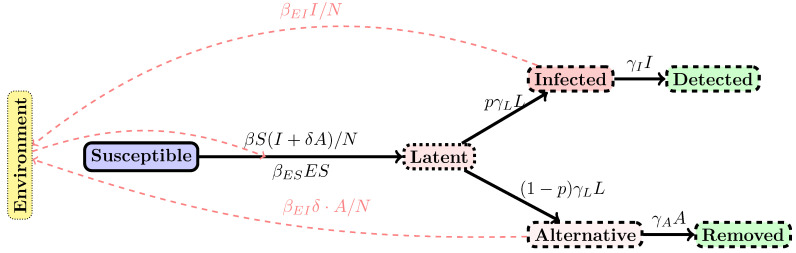
SLIADRE model: Schematic view of the SLIADRE model used in the simulations. Each compartment is indicated with a rectangular box and the flow of individuals or infection vectors with solid arrows. Dashed arrows indicate which compartment influences another.

**Figure 2 biology-10-00010-f002:**
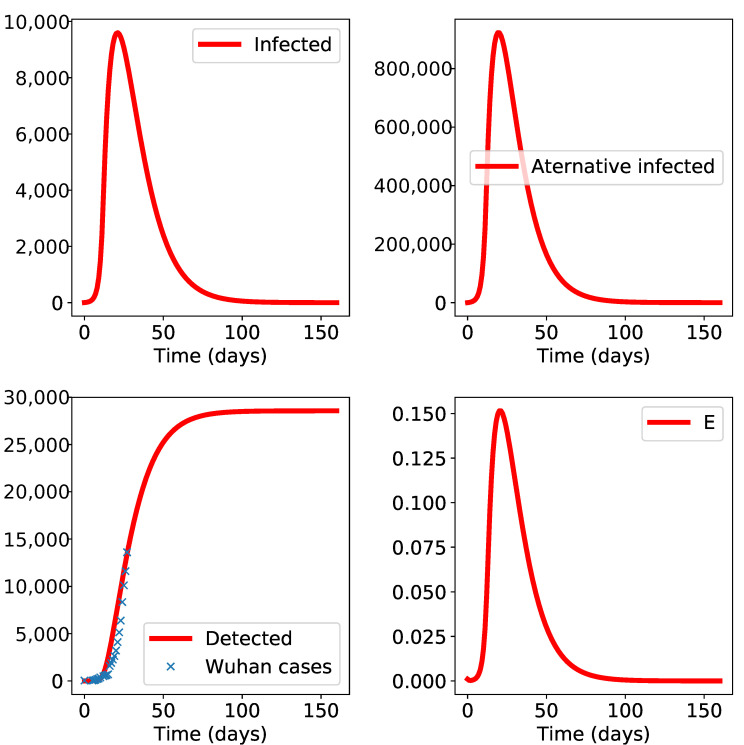
Results of the model for Wuhan city epidemic. The initial date $t_{0}$ is set to 11 January 2020 and the data available up to 9th February is used in this simulation. The class “E” is increasing which indicates important environmental propagation.

**Figure 3 biology-10-00010-f003:**
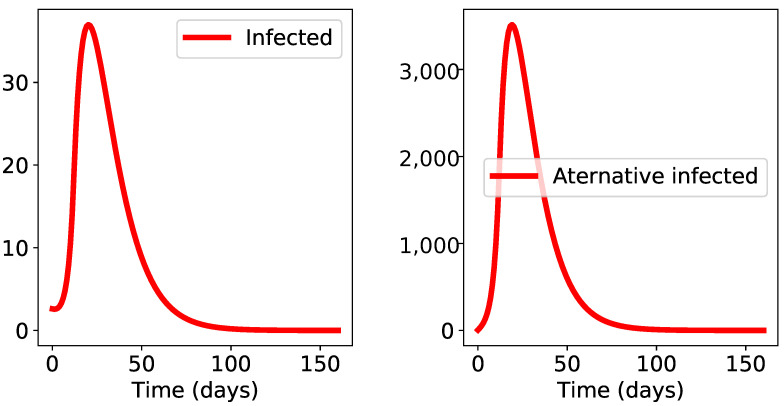
Results of the simulations when the environmental propagation route E-S is switched off. The classical propagation route alone induces a much smaller epidemic size than the actual data indicate.

**Figure 4 biology-10-00010-f004:**
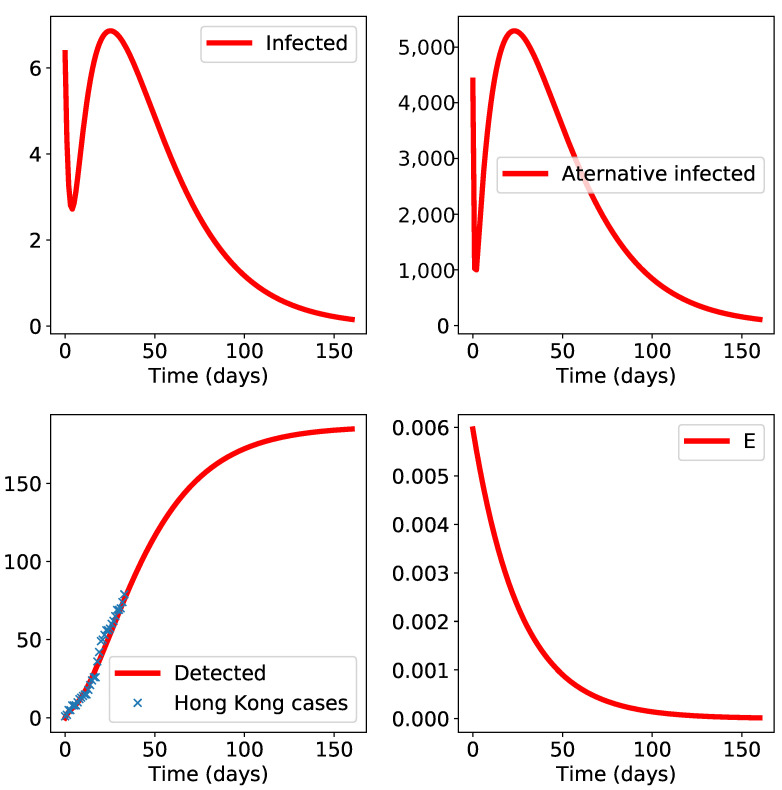
Simulation results for Hong Kong. The initial time $t_{0}$ is set to 22nd January. The class “E” is small and decreasing, which indicates negligible environmental propagation.

**Figure 5 biology-10-00010-f005:**
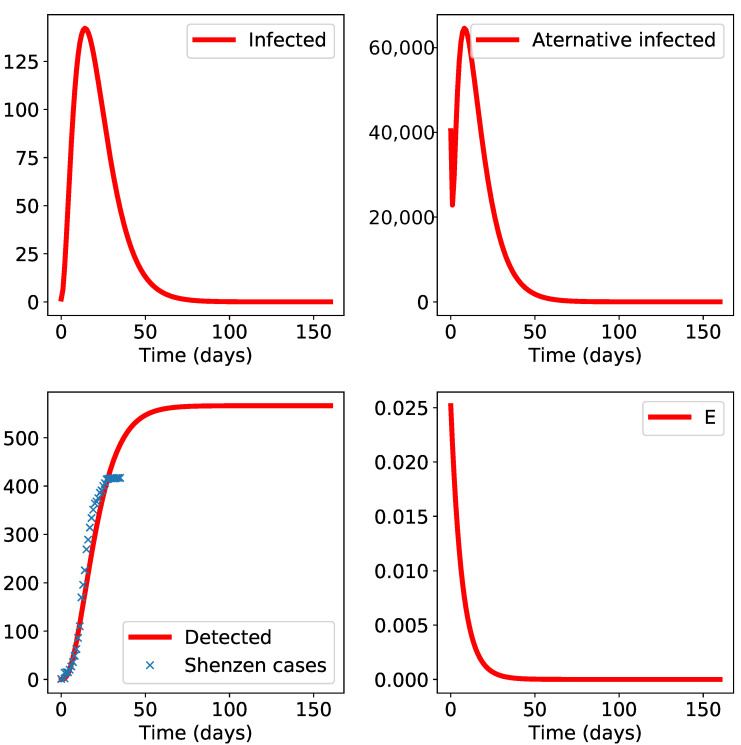
Results of the SLIADRE model for Shenzen. The initial time $t_{0}$ is set to January 19th. The apparent misfit at the end of the empirical data time range is due to the fact that the database was not updated for several days (thus the cumulative numbers will show a constant trend, in contradiction with the real epidemic behavior). The class “E” is small and decreasing which indicates negligible environmental propagation.

**Figure 6 biology-10-00010-f006:**
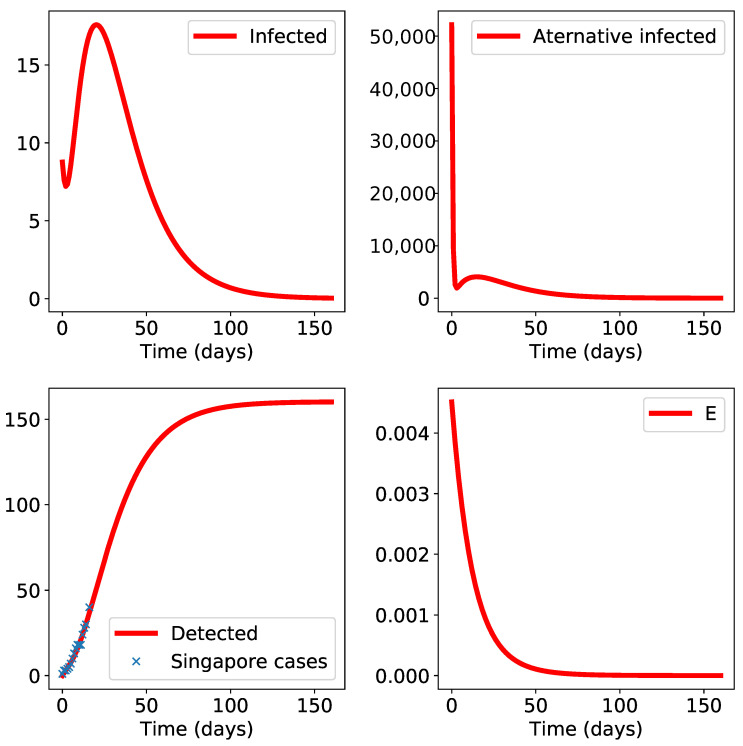
Simulation results for Singapore. The initial time $t_{0}$ is set to 23rd January. The class “E” is small and decreasing which indicates negligible environmental propagation.

## Data Availability

Data available in a publicly accessible repository that does not issue DOIs. Publicly available datasets were analyzed in this study. This data can be found here: http://wjw.hubei.gov.cn/fbjd/dtyw/, http://wjw.sz.gov.cn/yqxx/, https://www.coronavirus.gov.hk/eng/index.html#Updates_on_COVID-19_Situation.

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
