# Peer review of "A New Transmission Route for the Propagation of the SARS-CoV-2 Coronavirus"

_biology, 2020, doi:10.3390/biology10010010_

Round 1

Reviewer 1 Report

This manuscript is basically fine. However, one basic issue is not addressed, namely how to obtain the parameters (mainly the transition rates) from the available data. Which calibration routine is used?
This is relevant, since with increasing system size and ccmplexity the
calibration becomes more and more difficult and sensitive to errors.

Also, the figures should be enlarged and improved.

Reviewer 2 Report

The authors deals with an interesting topic about a possibility another transmission route of SARS-CoV-2. They point as a complementary route the oro-fecal transmission. I think this route can be important in sometimes depending on environmental conditions. 

However, major modifications are needed to be able to publish this article.

I would like the sentence from lines 52 and 53 on page 2 should be further developed. Why the selective pressure and the incubation time can be different?

Line 115 on page 7. Why the results of Singapore cases not are shown?

Line 196 on page 9. The bibliographic citation is incomplete.

Line 201 on page 9. The bibliographic citation is incomplete.

I miss the significance of the models fitted

I would like you explain more how relationship this new route with the transmission oro-fecal. Why this route and non another?

Round 2

Reviewer 2 Report

The authors have improved comprehension of the text, thank you